# ICENET: A Semantic Segmentation Deep Network for River Ice by Fusing Positional and Channel-Wise Attentive Features

**Xiuwei Zhang** [1,2,†], **Jiaojiao Jin** [1,2,†], **Zeze Lan** [1,2,*], **Chunjiang Li** [3], **Minhao Fan** [4], **Yafei Wang** [5], **Xin Yu** [3] and **Yanning Zhang** [1,2]

1   School of Computer Science and Technology, Northwestern Polytechnical University, Xi'an 710072, China; xwzhang@nwpu.edu.cn (X.Z.); jinjiaojiao@mail.nwpu.edu.cn (J.J.); ynzhang@nwpu.edu.cn (Y.Z.)
2   National Engineering Laboratory for Integrated Aero-Space-Ground-Ocean Big Data Application Technology, Xi'an 710072, China
3   Yellow River Institute of Hydraulic Research, Zhengzhou 450003, China; lichunjiang@hky.yrcc.gov.cn (C.L.); yuxin@hky.yrcc.gov.cn (X.Y.)
4   Hydrology Bureau of the Yellow River Conservancy Commission, Zhengzhou 450004, China; fanminhao@swj.yrcc.gov.cn
5   Ningxia–Inner Mongolia Hydrology and Water Resource Bureau, Baotou 014030, China; wangyafei1@swj.yrcc.gov.cn
*   Correspondence: msclab@mail.nwpu.edu.cn
†   These authors contributed equally to this work.

**Abstract:** River ice monitoring is of great significance for river management, ship navigation and ice hazard forecasting in cold-regions. Accurate ice segmentation is one most important pieces of technology in ice monitoring research. It can provide the prerequisite information for the calculation of ice cover density, drift ice speed, ice cover distribution, change detection and so on. Unmanned aerial vehicle (UAV) aerial photography has the advantages of higher spatial and temporal resolution. As UAV technology has become more popular and cheaper, it has been widely used in ice monitoring. So, we focused on river ice segmentation based on UAV remote sensing images. In this study, the NWPU_YRCC dataset was built for river ice segmentation, in which all images were captured by different UAVs in the region of the Yellow River, the most difficult river to manage in the world. To the best of our knowledge, this is the first public UAV image dataset for river ice segmentation. Meanwhile, a semantic segmentation deep convolution neural network by fusing positional and channel-wise attentive features is proposed for river ice semantic segmentation, named ICENET. Experiments demonstrated that the proposed ICENET outperforms the state-of-the-art methods, achieving a superior result on the NWPU_YRCC dataset.

**Keywords:** river ice; position attention; channel-wise attention; deep convolutional neural network; semantic segmentation

## 1. Introduction

Every winter and spring, river ice freeze-up and break-up are big events in cold regions. The presence of river ice cover can drastically affect various river flow characteristics and socioeconomic activities, such as water transportation, water supply and hydroelectric power [1]. Moreover, in the period of river ice freeze-up and break-up, a large jumbled accumulation of drift river ice can form an ice jam, which partially blocks a river channel, raises water levels and potentially causes flooding. This kind of flood by large rivers in cold regions is a well-known serious hazard called an ice jam flood.

It is potentially more destructive than open-water flooding and can produce much deeper and faster flooding. It also damage an economy by causing river-side industrial facilities such as hydro-electric generating stations to shut down and to interfere with ship transport [2–4].

So, river ice monitoring is often used to maintain navigation or predict ice jam related floods. To predict ice jam, accurate and timely information on the river ice cover and on the river ice distribution along the river channel is needed.

Imaging instruments with high spatial resolution are commonly-used for ice monitoring. Cooley [5] proposed a method to identify the spatial and temporal breakup patterns of river ice at large scales using MODIS satellite imagery. Chaouch [6] presented a technique to detect and monitor river ice using observations from the MODIS instrument onboard the Terra satellite. Chu [7] integrated use of different remote sensing data for river ice monitoring, using MOD09GQ data to characterize river ice phenology and Radarsat-2 data to classify breakup ice types. Ansari [8] developed an automated image processing algorithm to analyze the time series of terrestrial images, which was able to detect and quantify important river ice cover characteristics. Alfredsen [9] mapped complex ice formations using low cost drones and structure from motion. Kartoziia [10] assessed the icewedge polygon current state by means of UAV imagery analysis. In general, there are three kinds of monitoring method, including satellite-based monitoring [5–7], shore-based terrestrial monitoring [8] and UAV-based aerial monitoring [9,10]. Satellite-based ice monitoring has a large-scale observation and can estimate the river ice cover over large regions, but the satellite revisit period is often one or several days. Shore-based terrestrial monitoring of river ice can capture the information of river ice at any time. However, mounting an imaging instrument on the shore is not an easy thing, especially for a long and wide river running in the mountains. Compared with the above two methods, UAV-based aerial monitoring is an effective and complementary method to quickly detect or update the ice flood risk. It has the advantages of high spatial and temporal resolutions. So we focused on river ice monitoring based on UAV aerial image in this paper.

To the best of the authors' knowledge, there is no public UAV image dataset for river ice segmentation. To study an accurate semantic segmentation model for river ice monitoring, a UAV visible image dataset was built for river ice segmentation. The visible images in this dataset were captured by different UAVs at the Ningxia–Inner Mongolia reach of the Yellow River. The Yellow River, known as the most difficult river to manage in the world, spans 23 longitudes from east to west and 10 latitudes from north to south. There is a great disparity in topography and landform, and the flow amplitude varies greatly. The main stream and tributaries of the Yellow River have different degrees of ice in spring and winter. The ice on the Yellow River is very typical and diverse due to the influence of temperature, flow rate, geographical location and river morphology [11,12]. Therefore, the Yellow River was selected to study the river ice segmentation models.

In this paper, we focus on building a UAV image dataset for river ice segmentation and designing a novel network architecture which effectively exploits multilevel features for generating high-resolution predictions. The main contributions of this paper are summarized as follows:

- A UAV visible image dataset named NWPU_YRCC was built for river ice semantic segmentation. It contains 814 accurately annotated images, which cover typical images of river ice in different periods, with diverse appearance and captured from different flight height and views.
- We propose a novel network architecture with two branches for river ice segmentation, named ICENET. One branch adopts a shallower convolution architecture, to extract low-level position-attentive features. The other branch was designed to extract multiscale high-level channel-wise attentive features. The aggregation of low-level finer features and high-level semantic features can generate high-resolution predictions for river ice.
- The proposed network achieves the state-of-the-art performance on the NWPU_YRCC dataset: 88% mean intersection-over-union (IoU), compared with DeepLabV3, DenseASPP, PSPNet, RefineNet and BiseNet.

## 2. Related Work

### 2.1. Ice Segmentation

Efforts have been devoted to study ice segmentation based on remote sensing images. Most existing methods adopt satellite remote sensing images captured by different sensors, such as moderate-resolution imaging spectroradiometry (MODIS), advanced very high resolution radiometry (AVHRR), synthetic aperture radar (SAR) and so on. These methods fall into three groups: traditional threshold methods, methods based on machine learning and methods based on neural networks.

**Traditional threshold methods.** Selkowitz and Forster (2016) [13] classified pixels as ice or snow by calculating the normalized difference snow index (NDSI) using Landsat TM and ETM+ images to map persistent ice and snow cover (PISC) across the western U.S. automatically. Liu et al. (2016) [14] employed a straightforward threshold method on the visible and infrared satellite images to identify sea and freshwater ice and estimate ice concentration. Su et al. (2013) [15] proposed an effective approach of gray level co-occurrence matrix (GLCM) texture analysis based on the ratio-threshold segmentation for Bohai Sea ice extraction using MODIS 250 m imagery. Experiments showed that this method is more reliable for sea ice segmentation compared with the conventional threshold method. In addition, Engram et al. (2018) [16] adopted a threshold method on log-transformed data to discriminate bedfast ice and floating ice with the SAR imagery across Arctic Alaska. Along with the above methods, Beaton et al. (2019) [17] presented a calibrated thresholds approach to classifying pixels as snow/ice, mixed ice/water or open water using MODIS satellite imagery.

**Methods based on traditional machine learning.** Using the development of machine learning, Deng and Clausi (2005) [18] focused on a novel Markov random field (MRF) to segment SAR sea ice imagery, which used a function-based parameter to weigh the two components in a Markov random field (MRF). This achieved unsupervised segmentation of sea ice imagery. Dabboor and Geldsetzer (2013) [19] applied a supervised maximum likelihood (ML) classification approach to classify the river covers as first-year ice (FYI), multiyear ice (MYI) and open water (OW) using SAR imagery in the Canadian Arctic. Chu and Lindenschmidt (2016) [7] adopted the fuzzy k-means clustering method to classify the river covers as open water, intact sheet ice, smooth rubble ice and rough rubble ice with integration of MODIS and RADARSAT-2. Romanov (2017) [20] proposed a decision-tree approach to detect ice with AVHRR data.

**Methods based on neural networks.** In traditional machine learning techniques, most of the applied features need to be identified by a domain expert, in order to reduce the complexity of the data and make patterns more visible to learning algorithms. On the other hand a neural network, especially a deep neural network, has strong mapping and generalization abilities, which can self-organize, self-study and fit an arbitrary, nonlinear relationship between a dependent variable and independent variables without an accurate mathematical model. If sufficient and high-quality labeled data is available, a deep neural network can extract features more efficiently from data in an incremental manner. Karvonen (2004) [21] presented an approach based on pulse-coupled neural networks (PCNNs) for segmentation and classification of Baltic Sea ice SAR images. With the wide application of CNN, Wang et al. (2016) [22] used a basic, deep convolutional neural network (CNN) to estimate ice concentration using dual-pol SAR scenes collected during melting. Remarkably, Singh et al. (2019) [23] used some semantic segmentation models (e.g., UNet [24], SegNet [25], DeepLab [26] and DenseNet [27]) based on CNNs to categorize segment river ice images into water and two distinct types of ice (frazil ice and anchor ice). It provided fairly good results and increased in accuracy compared to previous methods using support vector machines (SVMs). This indicates a promising for future exploration of deep convolutional neural networks applied in ice detection and segmentation to some extent.

### 2.2. Semantic Segmentation Based on a Deep Convolutional Neural Network

Semantic segmentation is a fundamental task and has shown great potential in a number of applications, such as scene understanding, autonomous driving, video surveillance and so on.

Moreover, due to the demands of some practical tasks (e.g., land classification, change detection and so on), semantic segmentation is required in remote sensing technology. A fully convolutional network (FCN) [28] was the pioneering work to replace the full connection layer at the end of a classification model with a convolution layer. This brought in a new way of thinking and a solution for semantic segmentation. Recently, semantic segmentation models based on FCN have been constantly emerging. They are generally divided into four categories: encoder-decoder structure, dilated convolutions, spatial pyramid pooling and recurrent neural networks.

**Encoder-decoder architectures.** Encoder-decoder structure based on FCN was proposed to recover high resolution representations from low resolution or mid resolution representations. SegNet (2017) [25] adopted maximum indices in the pooling layer instead of the features directly, introducing more encoding information and improving the segmentation resolution. Similar to SegNet, U-net (2015) [24] had a more structured network structure, and a better result has been obtained by splicing the results of each layer of the encoder into the decoder. RefineNet (2017) [29] made a well-designed RefineNet module that integrates the high resolution features with the low resolution features in a stage-wise refinement manner by using three individual components: residual conv unit (RCU), multiresolution fusion and chained residual pooling. GCN (2017) [30] used a large convolution kernel and decomposed the convolution kernel of a large kxk into two, 1xk and kx1, to balance the accuracy contradiction between location and classification. Gao et al. (2019) [31] proposed a method to extract roads from optical satellite images using encoder-decoder architectures and a deep residual convolutional neural network. Fuentes-Pacheco et al. (2019) [32] presented a convolutional neural network with an encoder-decoder architecture to address the problem of fig plant segmentation. El Adoui et al. (2019) [33] proposed a different encoder and decoder CNN architectures to automate the breast tumor segmentation in dynamic-contrast-enhanced magnetic resonance imaging based on SegNet [25] and U-Net [24].

**Dilated convolution.** Unlike encoder-decoder, dilated convolution (2015) [34] introduced a new parameter into the convolution kernel, which defined the spacing between values of kernel. It was designed to increase the receptive field without reducing the spatial resolution. That work removed the last two pooling layers from the pretrained classification VGG (2014) [35] network and replaced the subsequent convolution layers with dilated convolution. DRN (2017) [36] studied gridding artifacts introduced by dilation and developed an approach to remove these artifacts. DeepLabV1 (2014) [37] used dilated convolution and a fully-connected conditional random field (CRF) based on VGG (2014) [35]. In the same way, dilated convolution was applied with the ResNet (2016) [38] in DeepLabV2 (2017) [26] and DeepLabV3 (2017) [39]. Fu et al. (2017) [40] improved the density of output class maps by introducing atrous convolution. DDCMN (2019) [41] is a network for semantic mapping, called the dense dilated convolutions merging network, used to recognize multiscale and complex shaped objects with similar colors and textures.

**Spatial pyramid pooling.** Spatial pyramid pooling was adopted to aggregate multiscale context information for better segmentation. PSPNet (2017) [42] is a pyramid pooling module used to ensemble multiscale information in different sub-regions. DeepLabV2 (2017) [26] and DeepLabV3 (2017) [39] use dilated spatial pyramid pooling (ASPP) to realize multiscale information for semantic context. And that specific method was to use parallel dilated convolution with different dilated rates, obtaining better segmentation results. However, another work deemed that the ASPP module in the scale-axis was not dense enough and the receptive field was not large enough. Therefore, DenseASPP (2018) [43] was proposed to connect a group of dilated convolutional layers in a dense way, obtaining a larger scale range. Chen et al. (2018) [44] combined a spatial pyramid pooling module and encode-decoder structure to encode multiscale contextual information and capture sharper object boundaries by recovering the spatial information gradually. He et al. (2019) [45] improved the performance of the road extraction network by integrating atrous spatial pyramid pooling (ASPP) with an encoder-decoder network.

**Recurrent neural networks.** Recurrent neural networks have been successfully applied for modeling long-temporal sequences. RNNs are able to exploit long-range dependencies and improve semantic segmentation accuracy successfully. Byeon (2015) [46] used two-dimensional long short term memory recurrent neural networks (2D LSTM networks) to address the problem of pixel-level segmentation. Inspired by the same recurrent neural network (ReNet) (2015) [47] architecture, Li (2016) [48] proposed a novel long short-term memorized context fusion (LSTM-CF) model for scene labeling. The DAG-RNNs (2016) [49] model was proposed to process DAG-structured data and effectively encode long-range contextual information. Shuai (2017) [50] built a recurrent neural network with a directed acyclic graph to model global contexts and improve semantic segmentation by linking pixel-level and local information together. These methods with recurrent neural networks can capture the global relationship implicitly. However, their effectiveness relies heavily on the learning outcome of the long-term memorization.

## 3. NWPU_YRCC River Ice Dataset

### 3.1. Motivation

Most studies on river ice segmentation are based on satellite remote sensing images. The related studies have been described in Section 2.1. Satellite imaging can map large scale scenes, but still faces the problems of a long transit period and low spatial resolution, and may suffer from cloud cover [51,52]. The UAV remote sensing platform has a high flexibility of use and can capture high spatial resolution images at low cost, because UAVs can fly at low altitudes above ground level [53]. With the wide application of UAVs, UAV aerial photography can be a complementary feasible approach for river ice regime observation, providing data with high temporal and spatial resolutions [54]. Deep learning techniques have been applied to a variety of practical applications over the past few years. To the best of our knowledge, there is no public UAV image dataset for river ice monitoring. To promote the study of river ice monitoring based on deep learning, a UAV visible image dataset was built for river ice segmentation. As mentioned in the first section, the ice on the Yellow River, especially in the Ningxia–Inner Mongolia reach, is very typical and diverse. Therefore, the Yellow River was selected to study the river ice segmentation. To build this dataset, four institutions devoted their efforts, including Northwestern Polytechnical University, the Yellow River Institute of Hydraulic Research, the Hydrology Bureau of the Yellow River Conservancy Commission and the Ningxia–Inner Mongolia Hydrology Bureau. The latter three are the subordinates of the Yellow River Conservancy Commission. Thus, it was named the NWPU_YRCC river ice dataset, short for the Northwestern Polytechnical University/Yellow River Conservancy Commission dataset.

The appearance of river ice varies a lot at different times; in order to cover different periods of river ice, we took a lot of aerial videos during November–March over four years. A total of 814 typical frames were carefully selected from these video sequences to form the dataset. All frames in this dataset were accurately annotated pixel by pixel by three classification labels, ice, water and shore. The details of this dataset are in the following two subsections.

Based on the NWPU_YRCC dataset, the problem of river ice extraction can be explored. Moreover, further analysis and studies based on ice extraction become feasible, such as ice cover density, ice cover distribution and change detection. All of them are very important for river ice monitoring.

### 3.2. Dataset Construction

**Collection.** The aerial images were captured at the Ningxia–Inner Mongolia reach of the Yellow River during November–March of each year from 2015 to 2019. In order to obtain an intuitive view and understanding of environments around image acquisition areas, we exhibit a map in Figure 1, in which the red ellipse reveals our study and imaging area. During the data collection, we used two different UAVs, a fixed wing UAV ASN216 with a Canon 5DS visible light camera and a DJI Inspire 1.

Some details about the UAVs and cameras are in Tables 1 and 2 respectively. The flight height of UAVs ranged from 30 to 500 m. There are 200 videos in total, ranging in length from 10 min to 50 min. Some scenes of experimental data collection are shown in Figure 2. The three pictures in the first row show the data collection procedure using a fixed wing UAV ASN216, including preparing the UAV, flying the UAV and the UAV's ground control station. The second row depicts some pictures of data collection utilizing DJI Inspire 1.

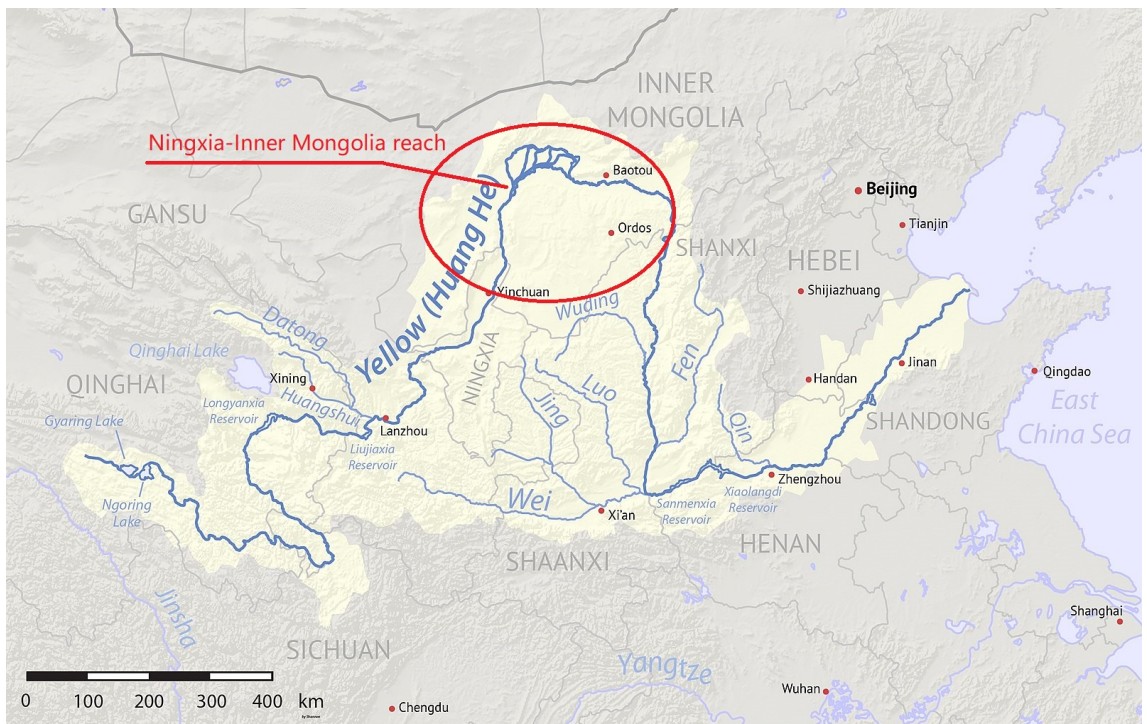

**Figure 1.** Study areas.

**Table 1.** Details of UAVs used for data collection.

| Parameters | ASN216 | DJI Inspire 1 |
|---|---|---|
| **Max Take-off Weight** | 30 kg | 3.4 kg |
| **Endurance** | ⩾4 h | ⩽18 min |
| **Max Speed** | 120 km/h | 22 m/s |

**Table 2.** Details of cameras used for data collection.

| Parameters | Canon 5DS | Camera on DJI Inspire 1 |
|---|---|---|
| **Sensor type** | CMOS | Exmor R CMOS |
| **Max Image resolution** | 8688*5792 | 4000*3000 |
| **Max Video resolution** | 1920*1080 | 4096*2160 |
| **Effective pixels** | 50.6 million pixels | 12.4 million pixels |

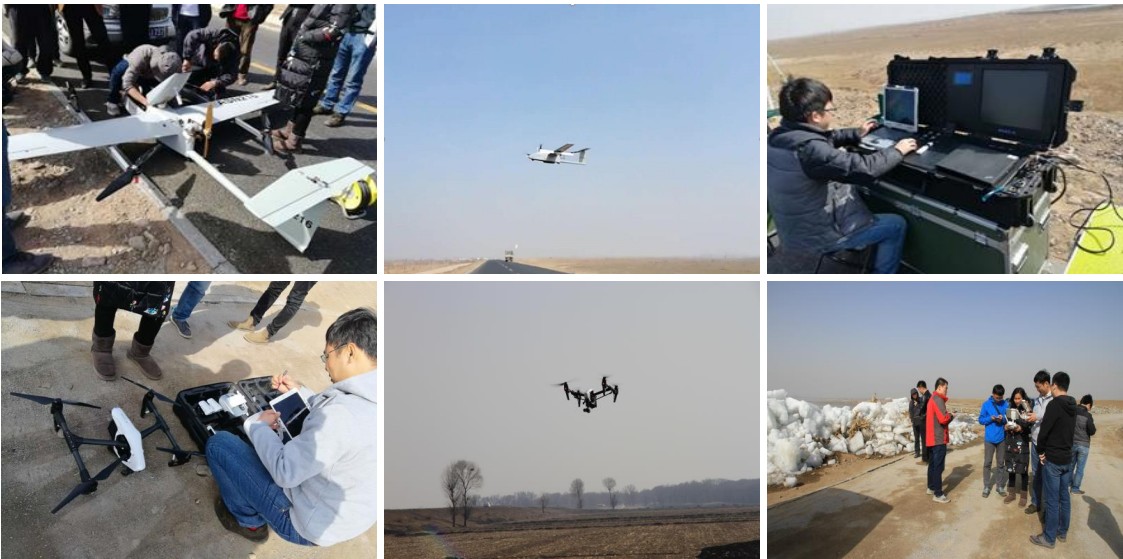

**Figure 2.** Experimental data collection.

**Manual annotation.** There are several annotation tools for semantic segmentation, such as Labelme, Image Labeler and so on. However, due to the irregularity of river ice, it is hard to mark a clear boundary between water and ice on the NWPU_YRCC dataset with these tools. Eventually, Photoshop software was adopted to label each pixel in the image as one of three categories, including ice, water and shore. However, there continued to be some problems. Other than three demanding kinds of pixel values, there were still other values in the annotation map. So the unexpected values were classified into three expected kinds using the least Euclidean distance metric. Some very small misclassified regions with only several pixels still existed. To ensure the integrity, an open morphology operation was carried out to correct them. This annotation job is very time-consuming. Finally, 814 typical images were carefully selected to be accurately annotated, and make up the NWPU_YRCC dataset. Note that the resolution of all annotated images is 1600 × 640. However, the resolutions of the collected images are diverse. As shown in Table 2, the maximum image resolution and video resolution of Canon 5DS on ASN216 are, respectively, 8688 × 5792 and 1920 × 1080, while the max image resolution and video resolution of camera on DJI Inspire 1 are 4000 × 3000 and 4096 × 2160. To ensure that the resolution of the input images in the deep convolution neural network was consistent, we finally cropped and resized the images to 1600 × 640. Furthermore, to depict the characteristics of the dataset, the ratio of ice pixels for each labeled image was statistically calculated and displayed in Table 3. Here, the ratio of ice pixels means the number of ice pixels to the number of river channel pixels (ice and water). It can depict ice density. From the Table 3, the dataset covers almost situations of differing ice density.

**Table 3.** The ratios of ice pixels in the labeled images.

| The Percentage of Ice Pixels | The Number of Images |
| :---: | :---: |
| 0–10% | 14 |
| 10–20% | 50 |
| 20–30% | 75 |
| 30–40% | 100 |
| 40–50% | 119 |
| 50–60% | 108 |
| 60–70% | 115 |
| 70–80% | 116 |
| 80–90% | 83 |
| 90–100% | 34 |

Figure 3 illustrates some typical images and their corresponding annotation maps. The first two rows present the images with ice of different size and color, and their annotation maps. The last two rows shows some sample images captured by different flight heights and view points and their annotation maps.

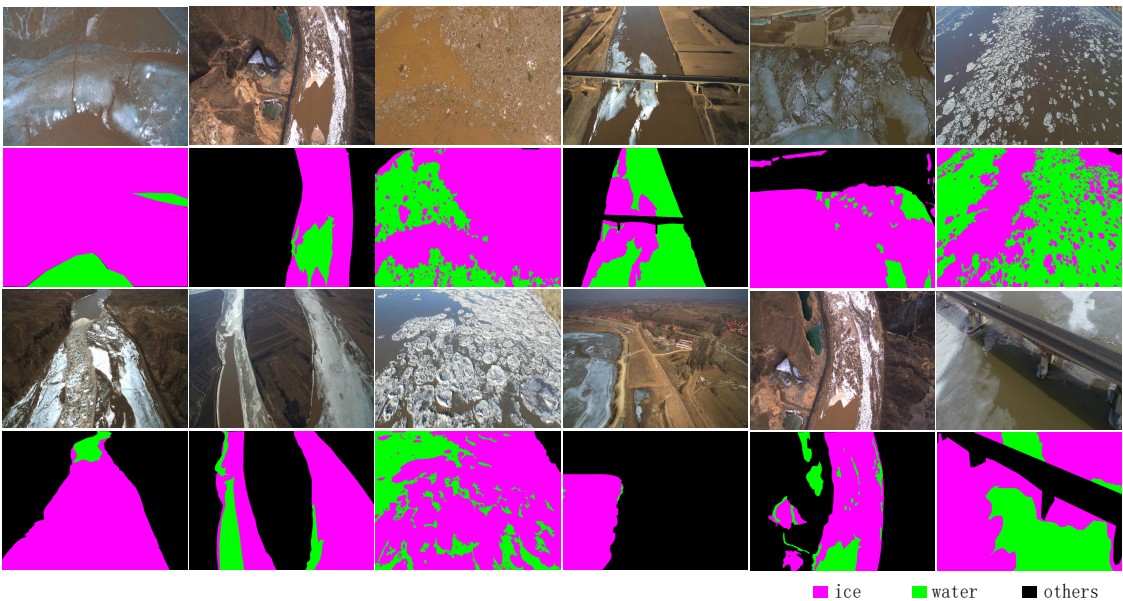

**Figure 3.** Some examples of image annotation.

*3.3. Analysis*

In summary, the NWPU_YRCC dataset has the following four characteristics.

**Different periods.** The captured images and videos can be divided into two periods: freeze-up and break-up according to different states of river ice. The appearance of river ice in these two periods varies greatly, including scale, color, texture and so on, as shown in the Figure 4. The first line shows some typical images of river ice in the freeze-up period. When the temperature drops to zero, small ice flowers are formed on the river. During the accumulation of small river ice, ice blocks of different sizes are formed on the river and along the shore. From the three images, we can see the shore ice and drift ice of different sizes. The second line shows some typical river ice images captured in the break-up period. As the temperature rises above zero (Condition 1) or crashes by the drift ice from upstream (Condition 2), the huge ice sheet starts to melt or breaks up into crushed ice of varying size. The left image shows the cracked ice caused by Condition 1. The other two images show the typical crushed ice scenes caused by both conditions. River ice jam occurs at the end of the accumulated ice in the middle image.

**Diverse appearance.** Compared with other rivers, the Yellow River has a more complex natural environment. Therefore, the river channel morphology and shore background are quite different. The river ice age can last from November in winter to march in the spring. And the color of river water and ice is often heavily influenced by different sediment content. Therefore, affected by temperature, channel conditions, flow power and sand-wind, the appearance of yellow river ice varies dramatically in scale, color, texture and shape. To make that more intuitive, some exemplary images are shown in Figure 5. The first line shows some images with clear water and drift ice blocks which are white or nearly white and of different sizes. The second line shows some images with big drift ice blocks which are in green or blue. The third line shows some images with ice and water seriously contaminated by sediment.

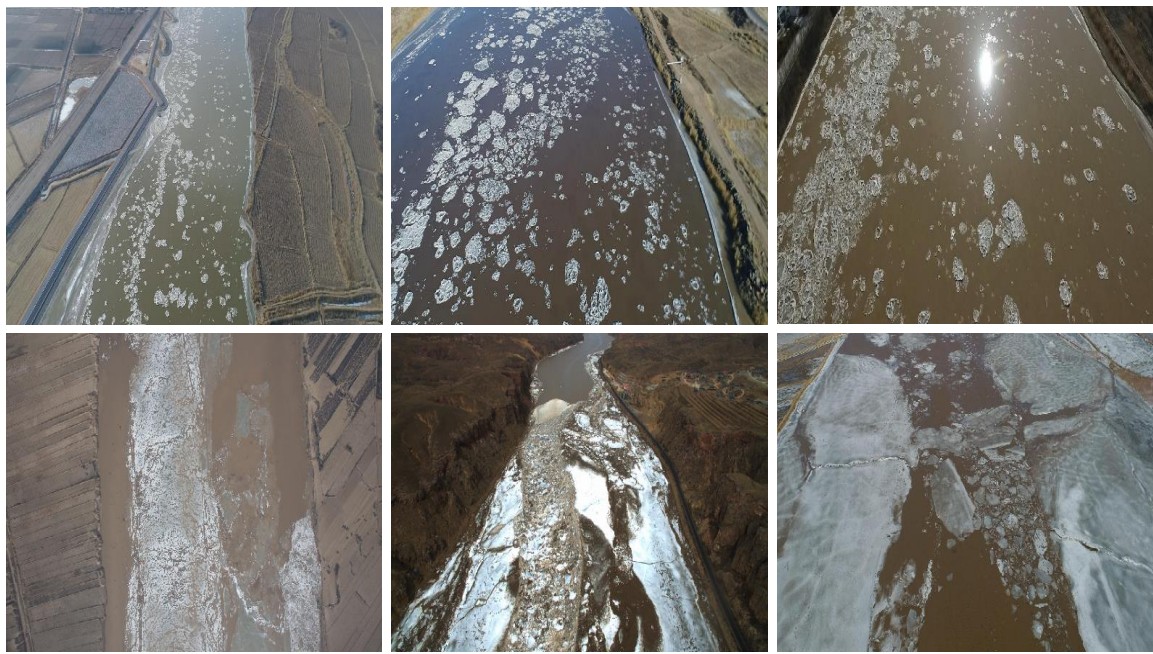

**Figure 4.** Typical river ice images in two different periods.

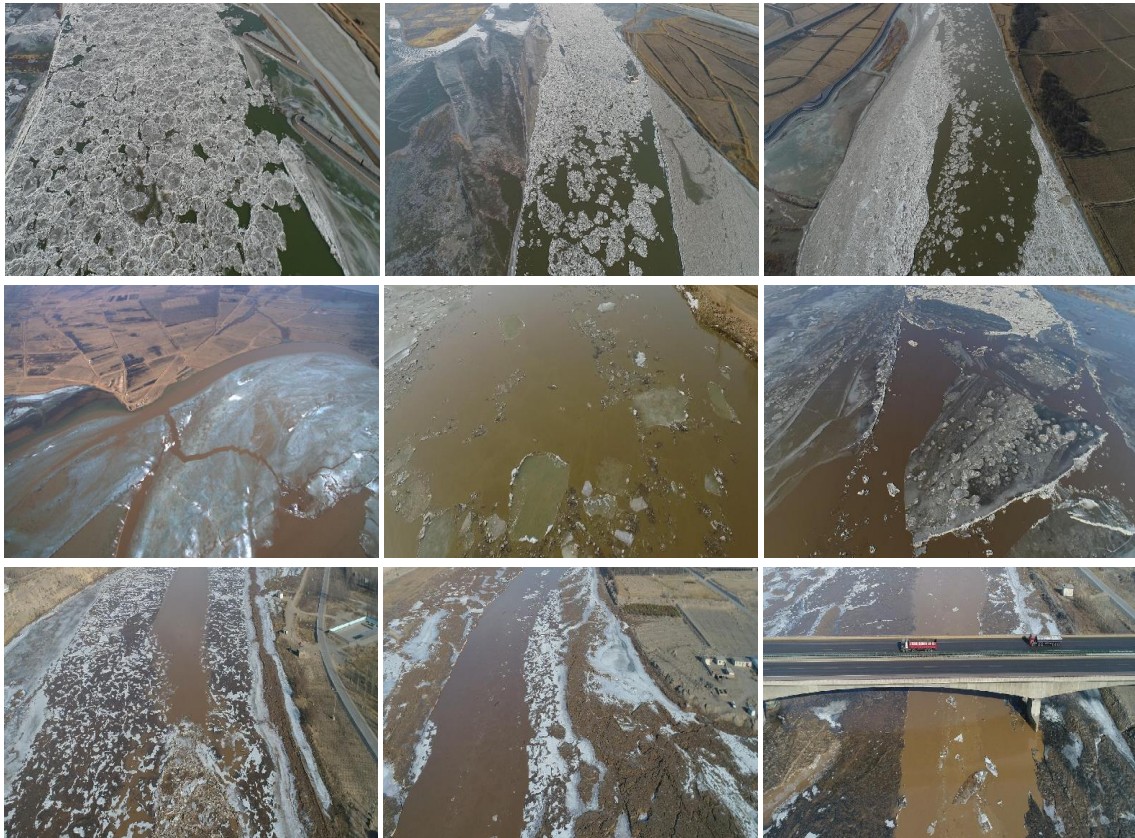

**Figure 5.** Example images of river ice with diverse appearance.

**Different flight heights and view points.** The flight height of the UAV and the view point of camera were different during the images/videos capture. The range of flight height was 30–600 m. Figure 6 presents some sample images captured from different fields of view and different angles in the dataset.

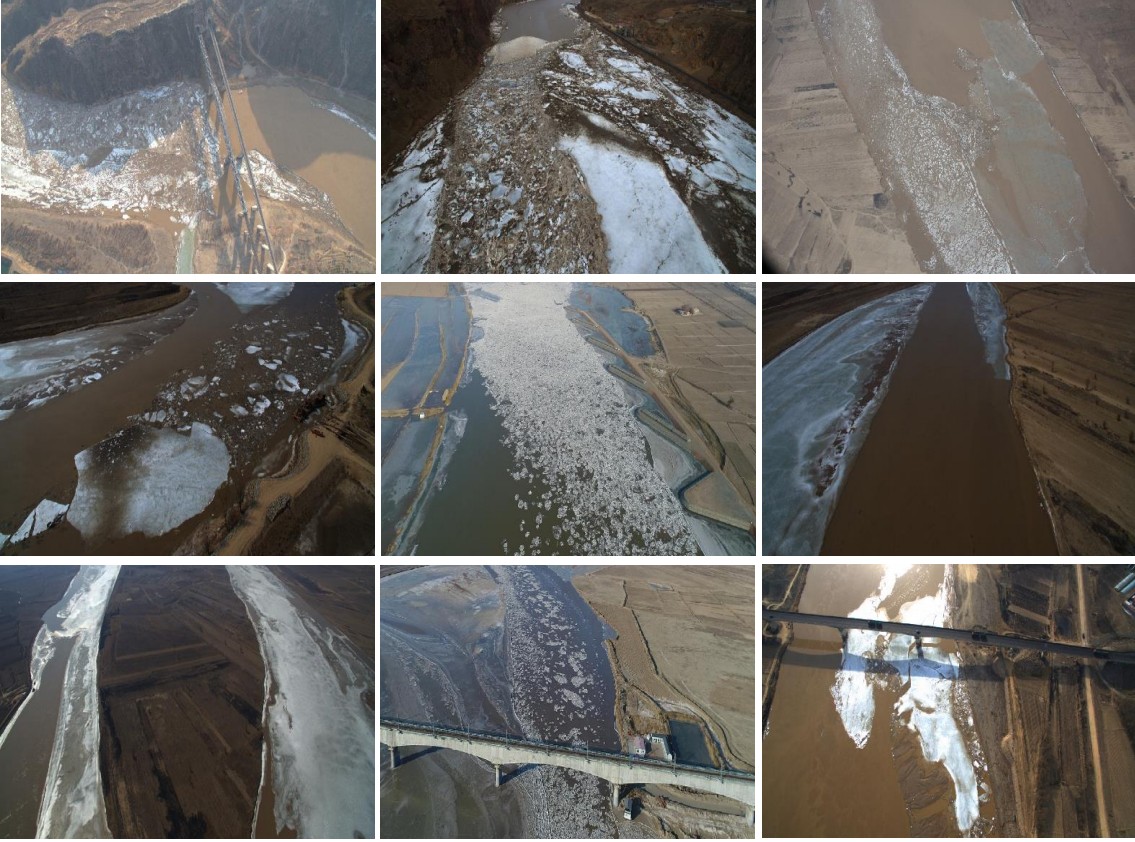

**Figure 6.** Example images captured from different flight heights and different view points.

As can be seen from the above characteristics of the NWPU_YRCC dataset, difficulties of segmentation on the dataset are two-fold: (1) The appearance of river ice in color, shape and texture varies a lot in different periods and different regions, so how to efficiently learn and integrate different level features to achieve accurate segmentation is the first challenge; (2) compared with the common objects in general scenarios—people, cars, etc.—on the PASCAL VOC or MS COCO dataset, the scale of river ice varies dramatically, so how to effectively segment ice at different scales, especially very little ice, is the second challenge.

## 4. Method

In this section, an overview of the proposed ICENET network is given first, which aggregates both the positional and channel-wise attentive features. Then, the four main parts of the proposed network: the multiscale feature aggregation, position attention, channel-wise attention and fusion module are presented in Sections 4.2–4.5 respectively.

### 4.1. Overview

To address the above two challenges of river ice segmentation, we designed a novel network of two branches shown in Figure 7. One branch adopts a deeper convolution architecture which fuses three channel-wise attention modules, aiming to extract multiscale features with high level semantic information.

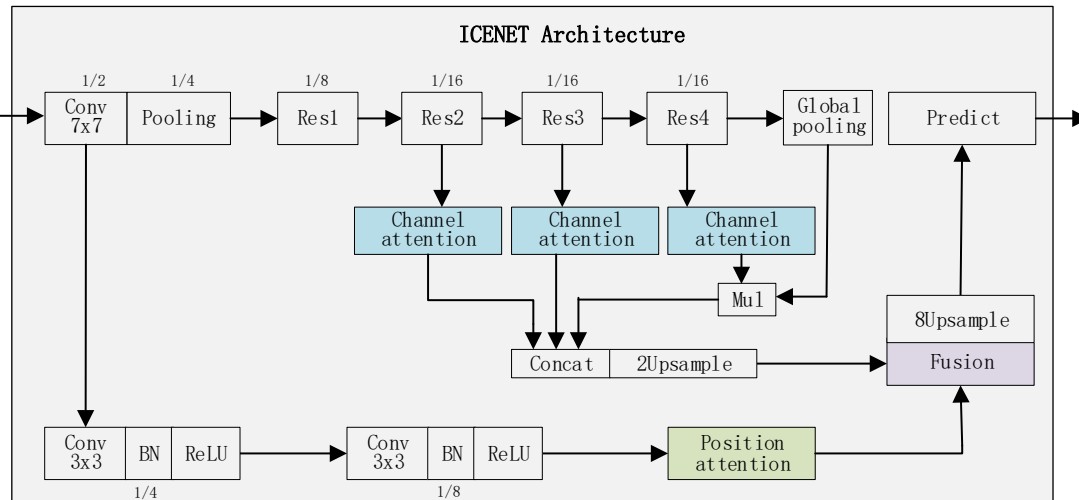

**Figure 7.** The ICENET architecture.

The other branch is designed as a much shallower convolution block which includes only two successive convolution layers (without a pooling layer) followed by a position attention module. As a result, the shallower branch can preserve high-resolution feature maps to capture small-scale targets. Then, the outputs of the two branches are combined to fuse both the positional and channel-wise attentive features to predict the final segmentation result.

## 4.2. Multiscale Feature Aggregation

Multiscale feature integration is an effective way to increase the segmentation accuracy of multiscale targets. Features of different levels have different representation capabilities. Low-level features can preserve high-resolution information and encode finer spatial information (e.g., edge, shape, and texture). While high-level features have strong semantic context owing to the larger receptive field, they tend to lose finer features due to the down-sampling operation. The key is to combine low-level and high-level features in a mutually beneficial way.

In the proposed model, the input image is fed into the ResNet-101 first, and the features are gained by a convolution layer with kernel size of seven and a stride of two. Then, the features are fed into two branches to obtain the detailed information and the semantic information, respectively.

The deep branch, e.g., the top one depicted in Figure 7, uses ResNet as the backbone. It contains four residual blocks, notated as Res1, Res2, Res3 and Res4. Only Res1 is an original residual block in ResNet. Convolutional strides of other three residual blocks (Res2, Res3 and Res4) are set to one. And a batch-normalization layer is added on top of Relu operation in Res4. Then, each of these three blocks are followed by a channel attention module. Furthermore, to get the global context information, the last channel attention module is weighted by a global contextual vector, which is generated by a global average pooling on the output feature maps of Res4. The outputs of the first two channel attention modules and the last weighted attention module are then concatenated and up-sampled twice to get multiscale semantic information.

The shallow branch, e.g., the bottom one shown in Figure 7, indicates that the feature map goes through two successive convolution layers with a kernel size of three and a stride of two. That makes the size of the output feature map 1/8 of the input image, aiming to gain high-resolution spatial information. Further, the feature maps are fed into a position attention module. Position attention can improve intraclass compact and semantic consistency. The output features from the shallow branch can preserve high-resolution finer information and better discriminability of small targets.

Finally, the feature maps from the two branches are aggregated to predict by a fusion module depicted by Figure 8.

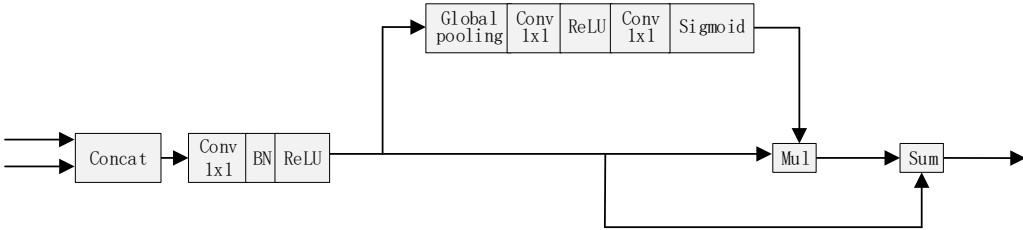

**Figure 8.** The fusion module.

### 4.3. Channel Attention

Channel attention can encode the relationship among feature channels by an attention vector, which is calculated among different channels of feature maps. It has been proven that channel attention can capture the contextual information and effectively improve network performance. Therefore, we designed a channel attention module for our model. The feature maps of Res2, Res3 and Res4 are all weighted by channel attention vectors to achieve high-level context information.

The structure of the channel attention module is presented in Figure 9. The input feature maps go through a global average pooling, and output a vector with the same size as the number of feature map channels. Then, a $1 \times 1$ convolution, followed by a batch normalization and a sigmoid function, are calculated to generate the attention vector. Finally the input feature maps are weighted by the attention vector, and element-wise added with themselves to produce the channel-wise attentive features. Through the channel attention module, the feature maps contributing to the segmentation task are emphasized, and the others are restrained.

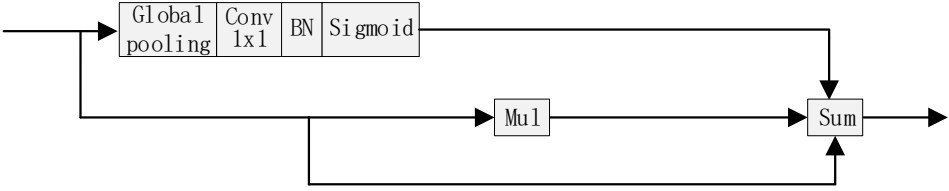

**Figure 9.** The channel attention module.

### 4.4. Position Attention

The position attention module is adopted in our method proposed in the dual attention network [55]. As shown in Figure 10, given a local feature map $\mathbf{A} \in \mathbb{R}^{C \times H \times W}$, the position attention firstly applies three convolution layers with 1 *times* 1 filters on $\mathbf{A}$ to generate three new feature maps $\mathbf{B}$, $\mathbf{C}$ and $\mathbf{D}$ respectively, where $\{\mathbf{B}, \mathbf{C}\} \in \mathbb{R}^{C \times H \times W}$. Then $\mathbf{B}$ and $\mathbf{C}$ are reshaped to $\mathbb{R}^{C \times N}$, where $N = H \times W$ is the number of pixels. After that, a matrix multiplication is performed between the transpose of $\mathbf{C}$ and $\mathbf{B}$, and a softmax layer is applied to obtain a spatial attention map $\mathbf{S} \in \mathbb{R}^{N \times N}$:

$$s_{ji} = \frac{e^{B_i \cdot C_j}}{\sum_{i=1}^{N} e^{B_i \cdot C_j}} \tag{1}$$

where $s_{ji}$ measures the $i$th position's impact on $j$th position. The more similar feature representations of the two position contributes to greater correlation between them.

Then, **D** is reshaped to $\mathbb{R}^{C \times N}$ and multiplied by the transpose of matrix **S**. The multiplication result is reshaped to $\mathbb{R}^{C \times H \times W}$. Finally, an element-wise addition operation is performed between the multiplication result and the original features **A** to get the final output $\mathbf{E} \in \mathbb{R}^{C \times H \times W}$ as follows:

$$E_j = \alpha \cdot \sum_{i=1}^{N} (s_{ji} D_i) + A_j \tag{2}$$

where $\alpha$ is initialized as 0 and gradually learns to assign more weight. Through the above process, similar features would be associated between two pixels regardless of their distance. The contextual information is added to the local feature **A** to augment the pixel-wise representation. Therefore, according to the position attention map, the feature map **E** has a global contextual view and aggregates contexts selectively.

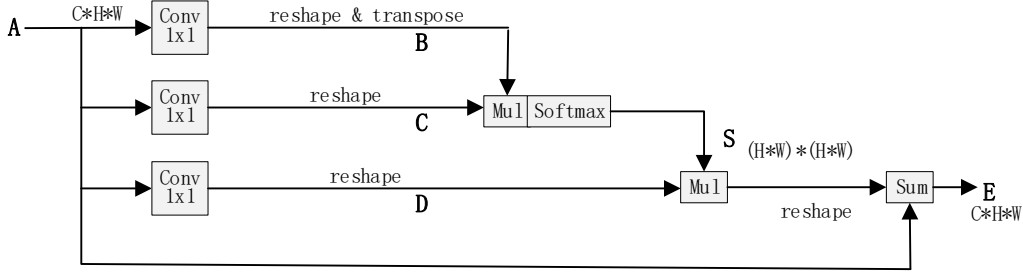

**Figure 10.** The position attention module.

### 4.5. Fusion Module

The features of the two branches are different in level of feature representation. Simply summing up these features is not sensible. Therefore, we adopt the feature fusion module used in [56] in our proposed architecture. As shown in Figure 8, the output features of the two branches are firstly concatenated; then, a convolution ($1 \times 1$ kernel size), batch normalization and ReLu unit are computed to balance the scales of the features. Then, a sub-module similar to channel attention module is calculated to generate a weight vector. This sub-model contains a global pooling, $1 \times 1$ convolution, Relu operation, $1 \times 1$ convolution and sigmoid unit sequentially. The concatenated and processed feature maps are weighted by the vector and summed with themselves to produce the fused features.

## 5. Experiments

In this section, the implementation details of the training process are presented first. Then, a series of ablation experiments are described. They were performed on the NWPU_YRCC dataset. Eventually, comparison results with some state-of-the-art methods are exhibited.

### 5.1. Implementation Details

There are 814 images that were used for annotation. The dataset was divided into 570 images for training, 82 images for validation and 244 images for testing. Note that the test set includes the validation set. We divided the dataset according to two typical periods: freeze-up and break-up. The numbers of images for two typical periods in different sets are shown in Table 4. We implemented our network based on the open source platform Tensorflow. The model was trained with RMSprop optimizer. The base learning rate was set to 0.0001 and the weight decay coefficient was 0.995. Batchsize was set to 1 and the training time was set to 200 epochs. In order to have a better visualization, the segmentation results were colorized. To be specific, the regions of ice, water and other are colored in purple, green and black respectively.

**Table 4.** The number of images for each period in the different sets.

| Set | Freeze-Up | Break-Up |
|---|---|---|
| training set | 295 | 275 |
| validation set | 50 | 32 |
| testing set | 146 | 98 |

*5.2. Evaluation*

Many evaluation criteria have been proposed, and some of them are frequently used to measure the accuracy of the system of semantic segmentation, such as pixel accuracy and IoU. In order to evaluate the performance of our model and other semantic segmentation algorithms objectively and scientifically, we need to use quantitative methods to calculate the performance indicators of segmented results. Therefore, we evaluated the performances of different models by measuring pixel accuracy (PA), mean intersection over union (MIoU) and F1-score, as shown in Equations (3)–(5) respectively. Additionally, ice was the key target in our study. To illustrate the performance improvement of the key target (ice), the respective IoUs of ice, water and other were also calculated in our experimental results.

We evaluated the performance on the assumption that there were a total of n + 1 categories (from 0 to n). These categories contained a void class or background (class 0) in the training and test dataset. In the evaluation metrics, $p_{ij}$ means the number of pixels that belong to class i but were predicted to be class j. In other words, $p_{ii}$ denotes the amount of pixels of class i inferred to belong to class i; namely, true positives. While when class i is regarded as positive and i is not equal to j, $p_{ji}$ and $p_{ij}$ represent the numbers of false positives and false negatives respectively.

**Pixel accuracy (PA).** Pixel accuracy, which is the simplest measure metric, refers to the ratio of the amount of properly classified pixels to the total number of pixels which are used to test the performance of algorithms. It is often denoted as follows:

$$PA = \frac{\sum_{i=0}^{n} p_{ii}}{\sum_{i=0}^{n} \sum_{j=0}^{n} p_{ij}}. \tag{3}$$

**Mean intersection over the union (MIoU).** The intersection-over-union (IoU), also known as the Jaccard Index, is one of the most commonly used metrics in semantic segmentation. The IoU is a very straightforward metric that is extremely effective. Simply put, the IoU is the area of overlap between the predicted segmentation and the ground truth divided by the area of union between the predicted segmentation and the ground truth. MIoU refers to the mean IoU of the image calculated by taking the IoU of each class and averaging them. Specific formula description are as follows:

$$MIoU = \frac{1}{n+1} \sum_{i=0}^{n} \frac{p_{ii}}{\sum_{j=0}^{n} p_{ij} + \sum_{j=0}^{n} p_{ji} - p_{ii}}. \tag{4}$$

Note that in our experimental results there is a light difference between the value of mIoU and the average of IoUs of ice, water and other. The reason is that some images in our test set only contained two classes of the three classes, ice, water and other.

**F1-score.** F1-score, taking the precision and recall rate of the classification result into account, refers to the weighted harmonic mean of precision and recall. It is usually devoted as follows:

$$F1 = \frac{1}{n+1} \sum_{i=0}^{n} \frac{2 \cdot precision_i \cdot recall_i}{precision_i + recall_i}, \tag{5}$$

where $precision_i$ and $recall_i$ refer to the precision and recall of class i, as shown in Equations (6) and (7) respectively. $precision_i$ is the ratio of the number of true positive pixels to the number of pixels which

are classified to class i. *recall*$_i$ refers to the ratio of the number of true positive pixels to the number of pixels which are class i in the dataset.

$$\text{precision}_i = \frac{p_{ii}}{p_{ii} + p_{ji}} \tag{6}$$

$$\text{recall}_i = \frac{p_{ii}}{p_{ii} + p_{ij}}. \tag{7}$$

*5.3. Ablation Study*

**Baseline.** The model shown in Figure 11 was used as the baseline. In the deep branch, the feature maps of Res4 are weighted by a contextual vector produced by a global average pooling, and then up-sampled twice as the output. We notate the combination of the block Res4, the global pooling layer and the multiplication operation as Res4*. The shallow branch only consists of two convolution layers. Then the features of the two branches need to be further aggregated and fused to generate the final prediction map.

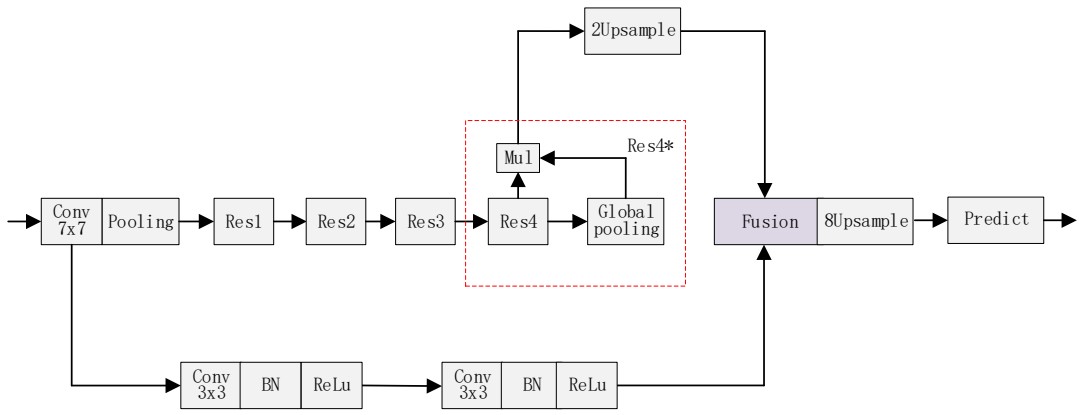

**Figure 11.** The base model.

**Ablation for fusion.** To evaluate the performance of different fusion strategies, based on the base model, we conducted experiments with different feature integration methods, and we present the results in Table 5. The setting fusion (Res3 + Res4*) means concatenating and up-sampling the feature maps of Res3 and Res4* as the output of the deep branch. Other settings have similar meanings. The experimental results demonstrate that fusing three blocks (Res2, Res3 and Res4*) is the most effective way. We name this model fusion3. The results of these methods are visualized in Figure 12.

**Table 5.** The performances of different feature integration methods.

| Method | IoU (%) | | | MIoU (%) | PA (%) | F1-Score(%) |
|---|---|---|---|---|---|---|
| | Ice | Water | Others | | | |
| baseline | 87.006 | 81.042 | 83.894 | 83.419 | 92.913 | 92.245 |
| fusion(Res3 + Res4*) | 88.481 | 81.040 | 86.460 | 84.915 | 94.365 | 93.988 |
| fusion(Res2 + Res4*) | 88.902 | 82.474 | 85.670 | 85.158 | 94.539 | 94.210 |
| **fusion3(Res2 + Res3 + Res4*)** | **91.136** | **83.909** | **88.296** | **87.535** | **95.813** | **95.709** |

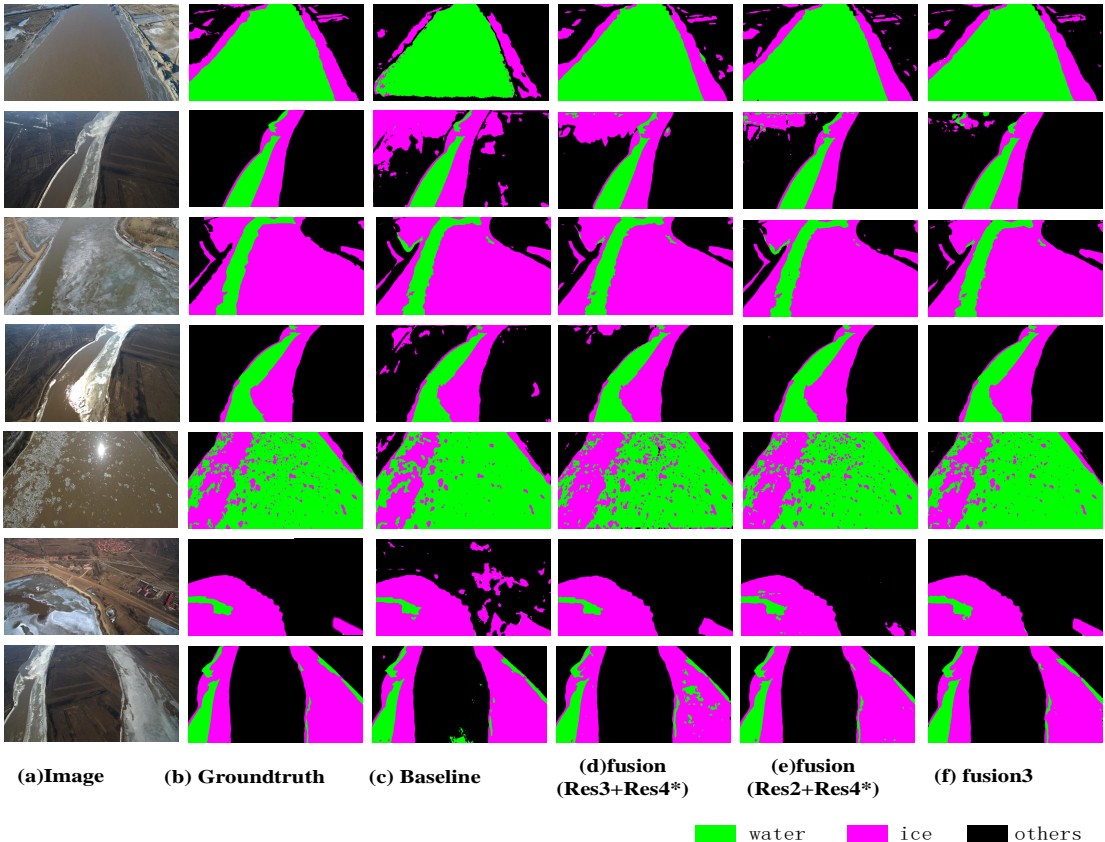

**Figure 12.** The results of different feature integration methods.

**Ablation for channel attention and position attention.** To verify the effectiveness of the channel attention module and the position attention module, experiments adding the channel attention and the position attention were performed based on the fusion3 model. It can be seen from Table 5 that each residual block of the fusion3 model has a certain contribution. Considering that channel attention can capture richer context information and is helpful for classification, we added channel attention on the tops of block Res2, Res3 and Res4, respectively. The experimental results are presented in Table 6. The notation CA2 is used to represent adding the channel attention on the top of the block Res2. And CA3 and CA4 have the similar meaning. It was observed that, compared with the fusion3 model, the effect become worse after adding the channel attention.

**Table 6.** The performances of channel attention and position attention.

| Method | IoU (%) | | | Mean IoU (%) | Pixel Accuracy (%) | F1-Score(%) |
|---|---|---|---|---|---|---|
| | Ice | Water | Others | | | |
| fusion3 | 91.136 | 83.909 | **88.296** | 87.535 | 95.813 | 95.709 |
| fusion3 + (CA2 + CA3 + CA4) | 90.287 | 83.356 | 87.077 | 86.693 | 95.096 | 94.744 |
| fusion3 + (CA2 + CA3 + CA4) + CA | 88.968 | 79.852 | 85.168 | 84.237 | 93.498 | 92.719 |
| **fusion3 + (CA2 + CA3 + CA4) + PA** | **91.583** | **84.891** | 88.253 | **88.112** | **95.932** | **95.814** |

Through analysis, we found that when features are multiplied by a residual channel attention, the features will be amplified. Relatively, the low-level features of the shallow branch may be restrained, only if adding the channel attention on the top of block Res2, Res3 and Res4. Then, we added channel attention to the shallow branch, but the performance dropped. Through further analysis, we found the channel attention emphasizes the relationship between different channels and is not suitable for low-level features. Position attention can improve intraclass compact and help to capture and position

small targets. Therefore, we use positional attention to replace the channel attention in the shallow branch, and achieved a great improvement.

In addition, the effects of the channel attention and the position attention are visualized in Figure 13. Some details and object boundaries are clearer with the attention module, such as the small ice blocks and the boundary between ice and water.

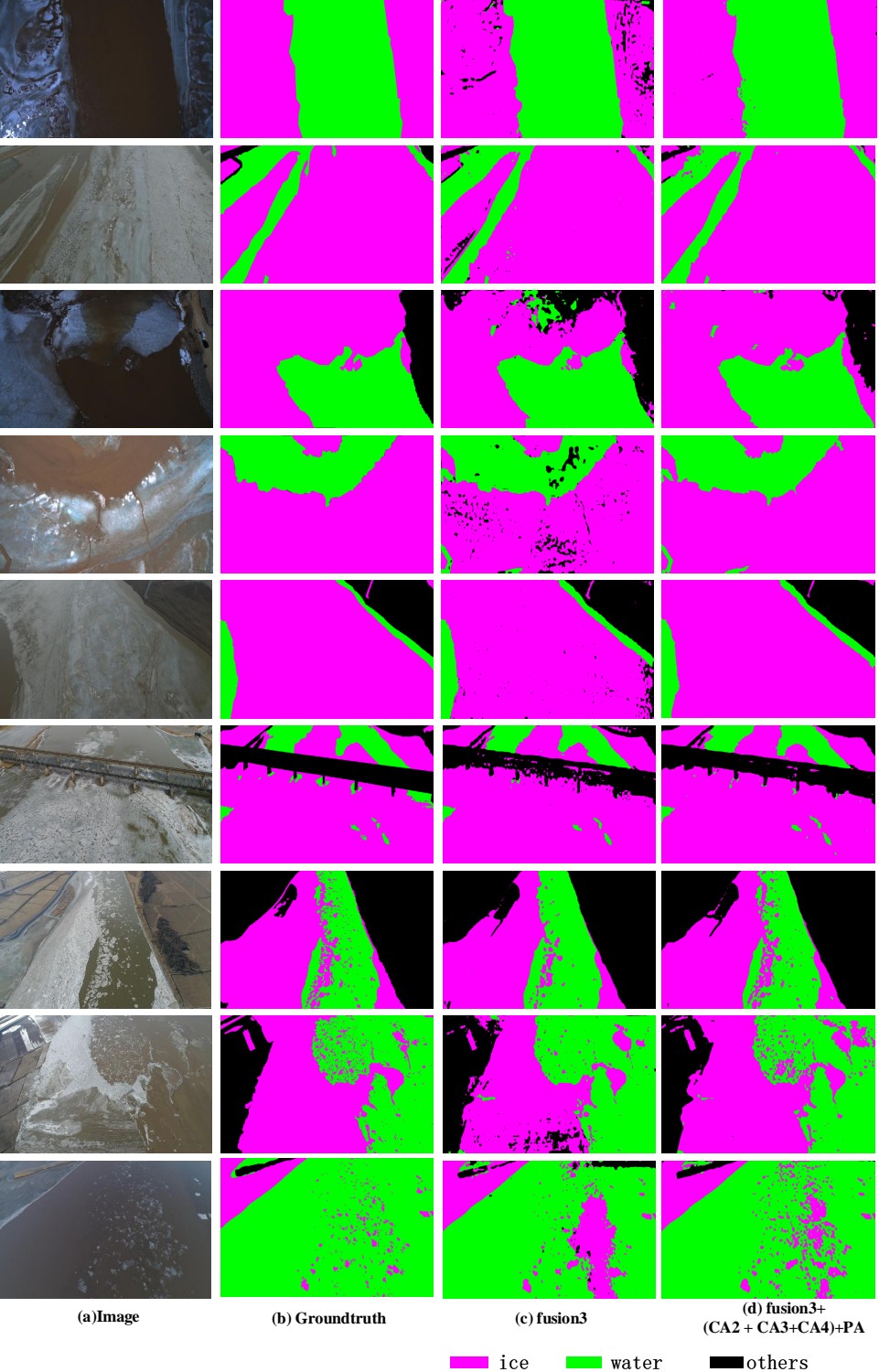

**Figure 13.** Visualization results of channel attention module and position attention module.

*5.4. Comparison with the State-of-the-Art*

We further make comparisons between the proposed method and the existing state-of-the-art methods. The code of the state-of-the-art models used in the paper was implemented in the Semantic-Segmentation-Suite project ( https://github.com/GeorgeSeif/Semantic-Segmentation-Suite) on the NWPU_YRCC dataset. Results are presented in Table 7. They indicate that the proposed method achieves significant improvements over other methods in terms of mean IoU. Figure 14 gives some visual comparison results of both the proposed method and other methods. They reveal that the proposed method acquires a satisfactory balance between detailed information and contextual information.

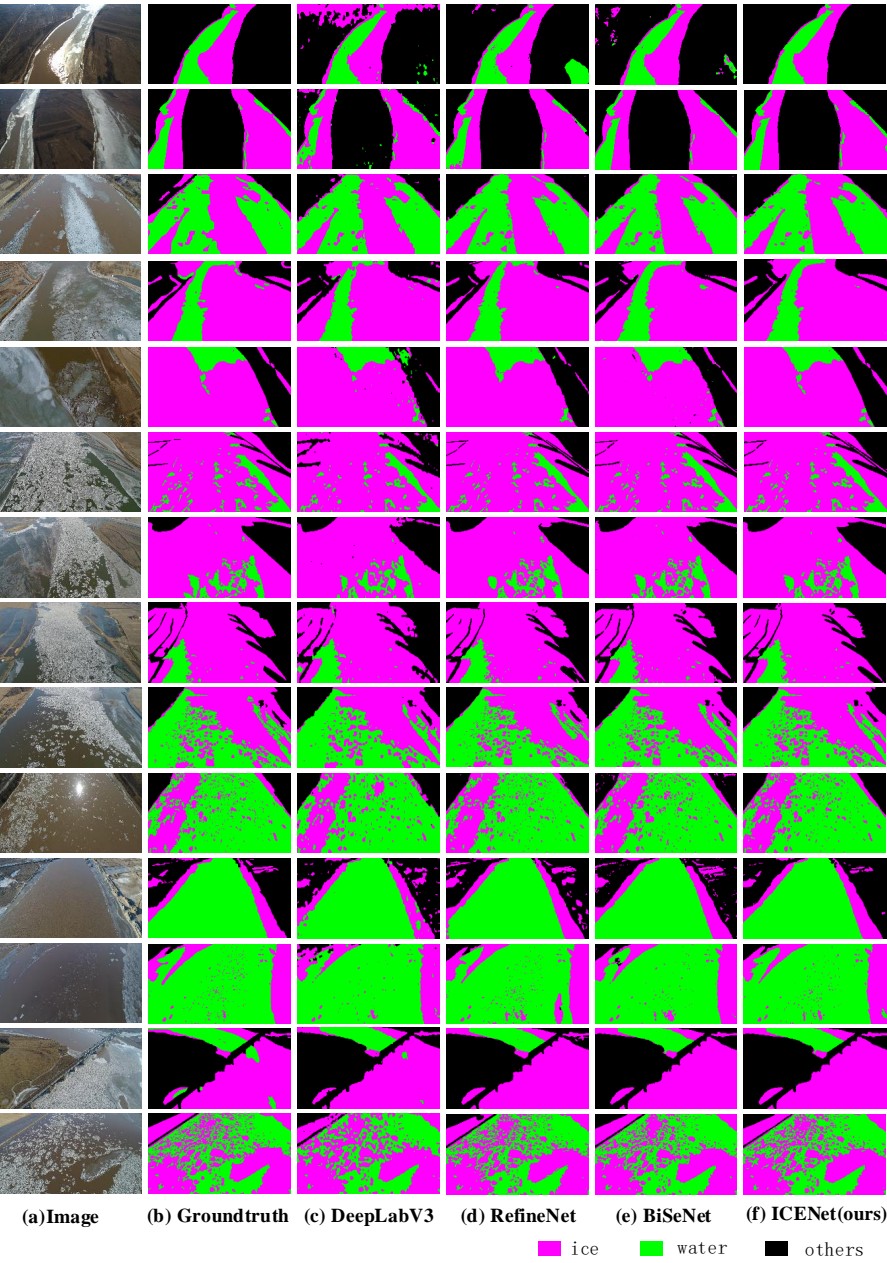

**Figure 14.** Visual comparison on NWPU_YRCC data.

**Table 7.** Comparison of the proposed method against other methods.

| Method | IoU (%) | | | Mean IoU (%) | Pixel Accuracy (%) | F1-Score(%) |
|---|---|---|---|---|---|---|
| | Ice | Water | Others | | | |
| DeepLabV3 [39] | 84.537 | 76.941 | 79.028 | 80.024 | 92.108 | 91.911 |
| DenseASPP [43] | 87.716 | 80.064 | 83.798 | 83.630 | 93.934 | 93.938 |
| PSPNet [42] | 88.196 | 81.483 | 83.774 | 84.374 | 93.966 | 93.707 |
| RefineNet [29] | 88.483 | 82.970 | 84.733 | 85.371 | 94.312 | 94.289 |
| BiseNet [56] | 89.301 | 83.464 | 87.814 | 86.497 | 95.058 | 94.820 |
| GCN [30] | 89.785 | 84.048 | 87.433 | 86.901 | 95.233 | 95.120 |
| **fusion3(ours)** | **91.136** | **83.909** | **88.296** | **87.535** | **95.913** | **95.732** |
| **ICENET(ours)** | **91.583** | **84.891** | **88.253** | **88.112** | **95.932** | **95.814** |

## 6. Conclusions

In this paper, we built a UAV visible image dataset named NWPU_YRCC for river ice semantic segmentation, aiming to apply deep neural network to assist river ice monitoring. Meanwhile, we proposed a novel network ICENET with two branches for river ice segmentation. One branch was designed as a deeper convolution architecture, to extract multiscale channel-wise attentive features with high level semantic information. The other branch adopts a much shallower convolution block, only two convolution layers followed by a position attention module, to preserve higher resolution feature maps. The outputs of the two branches are fused to predict the final segmentation result. The experiments showed that the model-based deep learning achieved good performance and the proposed method outperformed the state-of-the-art methods, achieving significant improvements on the NWPU_YRCC dataset. In addition, due to the high cost of labeling, the dataset is still not large enough to cover all realistic scenes. Therefore, we plan to use an active learning algorithm to enlarge our dataset in our future work.

**Author Contributions:** The idea of this research was conceived by X.Z., J.J. and Y.Z. Experimental data was captured and analyzed by C.L., Y.W. and M.F. The experiments were designed and carried out by X.Z., J.J. and Z.L. The manuscript was written by X.Z., J.J., Z.L. and M.F., and revised by Y.Z. and X.Y. All authors have read and agreed to the published version of the manuscript.

**Funding:** This research was funded by the National Natural Science Foundation of China under Grant 61971356, Grant 61801395 and Grant 6157240X5 and Grant U19B2037.

**Acknowledgments:** This research was supported by the National Natural Science Foundation of China (grant numbers 61971356, 61801395 and 6157240X5).

**Conflicts of Interest:** The authors declare no conflict of interest.

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
