# Peer review of "ICENET: A Semantic Segmentation Deep Network for River Ice by Fusing Positional and Channel-Wise Attentive Features"

_remotesensing, doi:10.3390/rs12020221_

Round 1
Reviewer 1 Report
In this paper, the Authors presented research which is of great interest for many reads in remote sensing. Although paper is well written and easy to follow, some corrections should be considered.
Section 3.1. Motivation
“Most studies on river ice segmentation are based on satellite remote sensing images. Satellite imaging can capture large scale scene, but faces the problems of long transit period and low spatial resolution”
Please refer to some current research in this area.
Please provide more detailed sensor/camera information (both for authors approach and referred research). It is well known that some satellites use a multispectral or even hyperspectral imaging camera. Should this multi-channel approach in UAV be of any benefit for the proposed research (unfortunately, a new multi-channel dataset should be required)?
Section 3.2. Dataset Construction
Drone shown on images is not DJI Phantom 3SE (DJI inspire?) Please provide more data about the used platform.
Are original images (4K or more resolution) resized before labeling and processing, or used in its original size? Is any image pre-processing performed before segmentation?
Line 218 Rive(r), a letter is missing
Line 229 and line 191, blank space between number and unit
Although well explained, evaluation methods should be more extensive. As this is not the usual classification scenario, and where misclassification/ false negative of ice pixels/ areas (may) cause some series problems. Please explain and expand.
As a reader, I would like to see performance off the classifier for ice pixels versus all others, as classification of ground and water body is nothing new neither interesting for the reads.
Also, provide some data about the ratio of ice pixels on labeled images, that should help us better understand segmentation performance.
There are small errors regarding the references:
Reference 1, missing volume, pages, etc.
Reference 15, a large part of the reference is missing
Please review other references and correct them according to the template.
re are no further suggestions (second round of review)
Author Response
Following your recommendation, we have revised our paper. Please refer to the appendix (reviewer1.pdf)

Reviewer 2 Report
ln 59: please explain "Mean IoU"
lns 80-99: neural networks are also a part of machine learning domain. Exceptional, but still - a part. I suggest to rephrase this part.
I suggest adding a comment about
General comments:
I suggest adding a comment about the advantage of the ICENET (or, in general, ANN) approach comparing to "traditional" machine learning techniques. Normally, classic spectral approach using those traditional methods could be a sufficient tool for this kind of task - assuming we have a typical spectral dataset. So, why CNN is better than, e.g. SVM?is it because of the lack of NIR bands? or different angles of acquisition? I suggest claryfying it in section 2.
Author Response
Following your recommendation, we have revised our paper. Please refer to the appendix (reviewer2.pdf)

Reviewer 3 Report
This paper presents a UAV image dataset named NWPU_YRCC for river ice semantic segmentation to apply deep neural network for river ice monitoring. Please consider the following comments.
Please provide proper references in the introduction section regarding ice jams, high-resolution image applications for river ice monitoring, characteristics of ice in Yellow River, etc. (Line 195) The UAV used in this study seems to be a DJI Inspire 1, not a Phantom 3. I would suggest providing maps revealing study area and imaging areas for a better understanding of environments around image acquisition areas. Although some descriptions are provided in the manuscript, please consider providing a detailed explanation for whole figures. Please provide examples of image annotation results. The authors mentioned that 814 images were used for annotation. But the dataset was divided into 570 images for training, 82 images for validation and 244 images for testing. Please provide a more detailed description of the image dataset regarding periods, imaging platforms, etc. Please add color legends in figures 10-12.Author Response
Following your recommendation, we have revised our paper. Please refer to the appendix (reviewer3.pdf)

Round 2
Reviewer 2 Report
The revised version includes my comments and suggestions.
This manuscript is a resubmission of an earlier submission. The following is a list of the peer review reports and author responses from that submission.